# Physico-Chemical Characterization of an Exocellular Sugars Tolerant Β-Glucosidase from Grape *Metschnikowia pulcherrima* Isolates

**DOI:** 10.3390/microorganisms11040964

**Published:** 2023-04-07

**Authors:** José Juan Mateo

**Affiliations:** Departamento de Microbiología y Ecología, Universidad de Valencia, 46100 Burjassot, Spain; jose.j.mateo@uv.es; Tel.: +34-96-354-3008

**Keywords:** *Metschnikowia pulcherrima*, grapes, characterization, β-glucosidase

## Abstract

A broad variety of microorganisms with useful characteristics in the field of biotechnology live on the surface of grapes; one of these microorganisms is *Metschnikowia pulcherrima*. This yeast secretes a β-glucosidase that can be used in fermentative processes to liberate aromatic compounds. In this work, the synthesis of an exocellular β-glucosidase has been demonstrated and the optimal conditions to maximize the enzyme’s effectiveness were determined. There was a maximum enzymatic activity at 28 °C and pH 4.5. Furthermore, the enzyme presents a great glucose and fructose tolerance, and to a lesser extent, ethanol tolerance. In addition, its activity was stimulated by calcium ions and low concentrations of ethanol and methanol. The impact of terpene content in wine was also determined. Because of these characteristics, β-glucosidase is a good candidate for use in enology.

## 1. Introduction

*Metschnikowia pulcherrima* is a non-*Saccharomyces* yeast present in various ecological niches, including the surface of grapes [1]. Morphologically, its shape is ovoid to ellipsoidal with a size of 2.5 μm × 4–10 μm. The diploid cells of this species propagate vegetatively by budding. Under certain anaerobic conditions, it can form pseudohyphae. It can also form one to two lance-shaped (acicular/threadlike) spores. Its colonies are cream-colored and produce a reddish-brown soluble pigment called pulcherrimina, which is characteristic of this species. It gives color to the colonies and diffuses it towards the medium [2]. Strains of *M. pulcherrima* can be identified using selective and differential media: they show the positive activity of the enzyme β-glucosidase, expression indicated using arbutin as a carbon source in agar plates, and proteolytic activity [3]. It grows well in media such as YPD (yeast extract peptone dextrose) or L-lysine. On the other hand, it shows very weak growth on nitrate agar [4].

*M. pulcherrima* is one of the non-*Saccharomyces* yeast species capable of expressing different extracellular hydrolytic enzymes, highlighting the following: amylase, cellulase, glucanase, β-glucosidase, β-lyase, lipase, lichenase, pectinase, protease, sulfite reductase, and xylanase [5,6]. β-glucosidases are β-d-glucoside glucohydrolases, capable of transferring a glycosyl group between nucleophilic oxygens, causing hydrolysis of β-glucosidic bonds between carbohydrate residues in various glycosides and producing short oligosaccharide chains and disaccharides. Under certain conditions, they can also synthesize the glycosidic bond [7]. Glycosides are molecules composed of a carbohydrate part, which can be a monosaccharide or a disaccharide, and a non-glycidic part, called aglycone, which are generally terpenes. However, they can also be phenolic acids, C-13 nor-isoprenoids, linear or cyclic alcohols, and possibly volatile phenols, which are all aroma precursors [8]. β-Glucosidases can be found in varied organisms, such as plants, fungi, and bacteria [9,10]. The intense aerobic glucosidase activity [3,11] promotes the liberation of varietal volatile compounds from the grape by hydrolyzing bound precursors. However, it is important to remember that the intensity of activity depends not only on the species but also on the strain [12].

In this work, the presence of *M. pulcherrima* on the surface of grapes was studied and isolates were identified using physiological and molecular techniques. The exocellular β-glucosidase produced by different *M. pulcherrima* isolates is characterized based on its biotechnological properties and its effect on wine as an aroma precursor.

## 2. Materials and Methods

### 2.1. Microbial Isolates

The grapes were harvested from Eastern Spain (Manchuela) in 2021 and 2022 when they were ripe. Grape juices were obtained and then stored at −20 °C. Samples were spread onto Malt Agar (20 g/L glucose, 20 g/L malt extract, 1 g/L mycological peptone, 15 g/L agar) and grown at 30 °C. Colonies suspected to be *M. pulcherrima* were isolated on Malt Agar and then identified.

### 2.2. Typing by Molecular Techniques

Pure cultures of isolates were grown on a Yeast Peptone Dextrose medium (20 g/L glucose, 20 g/L mycological peptone, 10 g/L yeast extract) at 30 °C for 24 h. Preparation of chromosomal DNA was performed using an Ultraclean Microbial DNA Isolation Kit (MoBio, Carlsbad, CA, USA) and stored at −20 °C for further assays. Isolates were analyzed by an RFLP analysis of the 5.8S-ITS rDNA region. PCR amplification was performed using Internal Transcribed Spacers ITS1 (5′-TCCGTAGGTGAA CCTGCGG-3′) and ITS4 (5′-TCCTCCGCTTATTGATAT GC-3′). Subsequently, PCR products were digested with the following restriction enzymes: HaeIII, CfoI, and HinfI and checked and measured by electrophoresis in 2.5% (*w*/*v*) agarose gel. For the sequence analysis of the D1/D2 domains of the 26S rDNA gene, PCR amplification was performed according to Kurtzman and Robnett [13] using NL1 (5′-GCA TATCAATAAGCGGAGGAAAAG-3′) and NL4 (5′-GGT CCGTGTTTCAAGACGG-3′) primers. The PCR product was purified using an UltraClean PCR Clean-Up kit (MoBio), and the sequence was determined by ABI Prism BigDye Terminator Cycle Sequence Ready Reaction Kit (Applied Biosystems, Stanford, TX, USA). The sequences were aligned using the BLAST search program, with complete or nearly complete 26S rDNA gene sequences retrieved from the EMBL nucleotide sequence data libraries (http://www.ncbi.nlm.nih.gov/BLAST/ (accessed on 20 May 2022)). Except where indicated, enzymes and molecular biology kits were purchased from Boehringer Mannheim GmbH, Mannheim, Germany.

### 2.3. Qualitative Screening of the β-Glucosidase Activity

An API ZYM strip (BioMerieux S.A.) was used to obtain preliminary detection of β-glucosidase. Plate screening for β-glucosidase activity was carried out on Esculin agar (2 g/L glucose, 1 g/L peptone, 1 g/L yeast extract, 0.3 g/L esculin, 0.01 g/L ferric ammonium citrate, and 15 g/L agar). Plates were incubated with 48-h yeast cultures at 30 °C for 3 days. The presence of the enzymatic activity was visualized as a dark halo around yeast growth.

### 2.4. Quantitative β-Glucosidase Assay

The selected isolates were inoculated into Yeast Peptone Dextrose at 30 °C for 48 h. An aliquot containing 5 × 10^6^ yeast cells was transferred to a sterile Eppendorf, centrifuged, and resuspended in 750 μL of 0.2 M citrate–0.1 M phosphate buffer (pH 6.0). Then, 250 μL of 5 mM pNP-glucoside suspended in the same buffer was added and the mixture was incubated at 50 °C for 90 min. The reaction was stopped by adding 1.0 mL of 0.2 M Na_2_CO_3_, and absorbance at 404 nm was measured.

### 2.5. Effect of Sugars and Ethanol on β-Glucosidase Activity

The effects of sugar on enzyme activity were conducted using sugar concentrations over a range of 0–200 g/L (glucose, fructose, and sucrose). The effect of ethanol and methanol on enzyme activity was conducted using alcohol concentration over a range of 0–20% (*v*/*v*). The activities were measured as described in Section 2.4.

### 2.6. Effect of Temperature and pH on β-Glucosidase Activity

The optimum pH was determined in 0.2 M citrate–0.1 M phosphate buffer covering a pH range from 2.2 to 8.0, at 50 °C for 90 min. The optimum temperature was measured from 4 °C to 70 °C for 90 min of incubation in the same buffer, at pH 6.0. The activities were measured as described in Section 2.4.

### 2.7. Effect of Metal Ions on β-Glucosidase Activity

The effect of different metal ions (KCl, ZnSO_4_, FeSO_4_, MgSO_4_, CuSO_4_, or CaCl_2_) on the β-glucosidase activity has been assayed at 2 mM and 10 mM. A sample with no addition has been used as blank. The activities were measured as described in Section 2.4.

### 2.8. Winemaking

The muscat juice, provided by Baronía de Turís winery (Valencia, Spain), was fermented by *S. cerevisiae* T73 (Lalvin). After 3 weeks, when less than 1.5 g/L of sugar remained, the wine was clarified by centrifugation. Samples were collected and then inoculated with *M. pulcherrima* isolates. They were grown in Yeast Peptone Dextrose at 30 °C for 48 h. After yeasts were centrifugated, they were resuspended in distilled water, added to 250 mL of Muscat wine at a final concentration of 5.0 × 10^6^ cfu/mL, and incubated at 18 °C for 14 days. The wine produced with *S. cerevisiae*, without adding *M. pulcherrima* isolates, was assayed and controlled.

### 2.9. Determination of Terpene Compounds Liberated from Wine Incubated with M. pulcherrima Strains

The isolation of terpenes was carried on C18 SPE columns (Waters). Volatile compounds were determined by using an Agilent 6890 N gas chromatograph-5973N mass detector system. All procedures were previously described [14].

### 2.10. Statistical Analyses

Statistical treatments (standard deviation, ANOVA) of results were performed using SPSS 24 software (IBM). The results were significant if *p* values were below 0.05. All assays were performed in triplicate.

## 3. Results and Discussion

### 3.1. Molecular Identification of Isolates

The RFLP of the PCR product of the ITS of all isolates suspected to belong to the *M. pulcherrima* species produced a characteristic band profile for each isolate. These were compared with data recorded in the Yeast-ID database at the Colección Española de Cultivos Tipo (CECT) web page, but no match was recorded. Subsequently, the D1/D2 region of genomic DNA was amplificated and identified by comparing sequences using the NCBI Blast program. The blast results showed that the D1/D2 gene sequence of all isolates had the highest similarity (99%) to that of *M. pulcherrima*.

### 3.2. Qualitative and Quantitative Detection of β-Glucosidase

The API ZYM strip is a commercially available, rapid enzyme testing system that determines the enzymatic profile of all *M. pulcherrima* isolates obtained in this work. Results obtained from these strips have shown that not all *M. pulcherrima* isolates produce an exocellular enzyme with β-glucosidase activity. To confirm these results, all isolates were cultured on Esculin Agar plates. Based on the darkness of the halo obtained, five isolates were selected for the next assays (isolates named Mp19, Mp20, Mp22, Mp24, and Mp26). Table 1 shows results for the quantitative detection of β-glucosidase activity in selected isolates, following the procedure described in Section 2.4.

### 3.3. Effect of pH and Temperature on β-Glucosidase Activity

Figure 1a shows the influence of pH on β-glucosidase activity. Maximum enzymatic activity was observed at a pH of 4.5, which drops below 50% at neutral pH, where most of the yeasts present an activity of around 30%. For most of the isolates, the enzyme still showed activity (more than 20%) at a very acidic pH.

Figure 1b shows the effect of temperature on enzyme activity. It presents its optimum temperature at 28 °C, even having considerable activity at 50 °C that goes from 50% to 90%. It should be noted that at 4 °C most isolates exceeded 25% activity.

β-Glucosidase from *M. pulcherrima* has its optimum activity at 28 °C and pH 4.5, which would suggest that the biological activity of this enzyme is optimized to the conditions in which it lives. This occurs in other species, such as the PC-2 strain of *Orpinomyces* sp., which has a β-glucosidase optimized to work in the rumen conditions of some animals, which is the habitat of the fungus [15]. As *M. pulcherrima* colonizes the surface of fruits, especially berries, and isolates used in this study were obtained from grapes, it would be expected that the optimum values for growing *Vitis vinifera* and *M. pulcherrima* would be the same [16]. Regarding the pH, these enzymes are usually acidic [17], which makes sense, since carboxyl groups are found in the catalytic domain. Therefore, the acidity would help to transfer the proton to the O-glycosidic bond and to protonate the carboxyl groups that act as a base, facilitating the hydrolytic activity of the enzyme. Some authors propose an optimum pH for β-glucosidase of 6.5–7.0 [18], while others report similar values to our findings [19]. The discrepancies can be attributed to the methodology used for the enzyme assay. The optimal pH for β-glucosidase activity in *Sporothrix schenckii* was 5.0 [20], which is similar to that described for the β-glucosidase of *Aspergillus* species [21,22]. The *S. schenckii* enzyme remained stable in the pH range of 5.0–8.0, retaining more than 85% activity, although it is sensitive to a pH below 4.0, losing its activity at a pH of 3.0 [20].

The temperature where more photosynthesis produces *Vitis vinifera*, the plant that produces grapes, is around 30 °C [23], approximately coinciding with the optimum of the β-glucosidase from *M. pulcherrima* [24]. Such optimum temperatures are within the range of temperatures determined by other authors for these enzymes [25,26].

### 3.4. Effect of Sugars on β-Glucosidase Activity

The effect of different sugars, specifically glucose, fructose, and sucrose on β-glucosidase activity was studied. Glucose (Figure 2a) showed no effect at low concentrations, but the activity slowly decreased as the glucose concentration increased. It is remarkable that at very high concentrations (200 mM) the activity remained above 40% of the maximum, with some isolates reaching more than 70%.

Low concentrations of fructose, from 1 to 10 mM, activate the enzyme but the higher content of this sugar slightly inhibits it, so that enzymatic activity was higher than 70% of maximum at a concentration of 200 mM. (Figure 2b).

Sucrose (Figure 2c) produces a stimulating effect at low concentrations and as the concentration of this sugar increased, the activity decreased slightly, leaving an activity greater than 30% at a concentration of 200 mM.

β-Glucosidase from *M. pulcherrima* presents an activity close to 100% in glucose concentrations generally found in wines [27]. In addition, the decrease in enzymatic activity is very gradual, with isolates reaching 70% activity in concentrations of 200 mM glucose. This means that β-glucosidase is glucose tolerant, which is its final product, and represents a very interesting feature in all possible applications of the enzyme since there are many β-glucosidases in various organisms that are inhibited by glucose [28,29,30].

In the β-glucosidases of Family 1, the most abundant and best studied, the catalytic domain is divided into three regions: the glycosyl group binding region, the aglycone binding region, and the gatekeeper region. This last region has residues that can help maintain the substrate and product dynamics within the catalytic domain, thus playing an essential role in glucose tolerance. However, this region is not as well preserved [31]. The gatekeeper region would attract glucose so that it does not accumulate in the regions where hydrolysis occurs. It has been observed that in some mutants, glucose binds to sites external to the entry channel to the catalytic region [32]. However, it has been shown that in mutants without this region, competitive inhibition by glucose occurs [33].

Some β-glucosidases are potentiated at low glucose concentrations. Some residues have polar interactions with a glucose molecule which creates a hydrogen bond between the structure formed with the carboxylic group of the catalytic domain. This indicates that both the carboxyl group and the substrate are in a position that decreases the amount of energy necessary to initiate the nucleophilic attack [32]. The β-glucosidase of *M. pulcherrima* does not present this stimulation by glucose at low concentrations, but it does occur by fructose, so it can be assumed that it is produced by the same mechanism.

In the case of sucrose, a disaccharide formed by glucose and fructose at low concentrations stimulates the enzyme, probably because it binds the fructose that forms this sugar allosterically, as occurs in fructose alone. However, at high concentrations of 40 mM, the glucose that forms sucrose binds to residues in the gatekeeper region. Furthermore, since it is a disaccharide and, therefore, larger than glucose, it hinders the transit of substrate into the catalytic domain, causing inhibition.

### 3.5. Effect of Alcohols on β-Glucosidase Activity

Regarding alcohols, ethanol increased β-glucosidase activity (Figure 3a) at low concentrations of up to 1% (*v*/*v*). This fact may be due to the slight modification of the protein structure due to hydrogen bonding caused by the presence of small amounts of ethanol. The activity then decreased drastically, yet at 200 mM there was still an activity greater than 30% and similar to that detected without the addition of ethanol, probably due to denaturation caused by alcohol. In the case of methanol (Figure 3b), an interesting result was obtained since two peaks of activity were perceived, one at a concentration of 0.5 mM and the other at 200 mM when a result like that of ethanol could be expected.

Ethanol has a similar effect, stimulating at low concentrations and inhibiting at high concentrations. This compound could act as a glycosyl acceptor, substituting water, and producing a greater β-glucosyltransferase activity of the enzyme. At higher concentrations, activity could be inhibited by denaturation or conformational changes of the enzyme by the apolar medium [34]. One drawback is that non-*Saccharomyces* yeasts do not usually tolerate conditions greater than 4–5% (*v*/*v*) of ethanol, so they disappear during the alcoholic fermentation of the wine [35]. However, the β-glucosidase of *M. pulcherrima* can continue with considerable activity at higher ethanol concentrations; so, it can be used during wine fermentations to release glucosidic-bound volatile compounds.

At low concentrations, methanol showed the effect observed for ethanol and at higher concentrations, the activity of the enzyme decreased up to 2% (*v*/*v*) of methanol. This is most likely due to the same phenomenon that occurs with ethanol. However, from 2% (*v*/*v*) of methanol, the enzymatic activity begins to rise.

These results make sense, as it has already been described in other yeast species with low fermentation capacities, such as *Candida wickerhamii*, *Pichia anomala,* and *C. molischiana*, that their β-glucosidases are more tolerant to more hostile conditions (low pH, low temperature, high amounts of ethanol or sugars) [8].

### 3.6. Effect of Chemicals on β-Glucosidase Activity

The effect of various cations on the β-glucosidase activity was also assayed (Table 2). The addition of Ca^2+^ highly increased β-glucosidase activity, particularly at 10 mM. These results could suggest the presence of acidic amino acids (Glu or Asp) in the active center of the enzyme which could be stabilized by this cation. A monovalent cation, which can only bind to a negative charge, would be unable to stabilize the protein structure. Among the divalent cations, it is observed that the greatest activity is obtained with the ion that has a greater covalent radius, which may be related to the separation that exists between the negative charges produced by the amino acids in the active center of the enzyme.

Given the purpose of the enzyme, it is not surprising that synergies occur between molecules present in grapes to enhance the degradation of cell walls. As previously mentioned, β-glucosidase is the last enzyme responsible for the degradation of cellulose, the second major polysaccharide in the cell walls of grapes [36]. However, 60% of the calcium in plant cells is found in the cell wall since it mediates cell adhesions [37]. Therefore, the calcium that would be released by breaking the cell walls could produce a conformational change in the enzyme, enhancing the degradation of cellulose to produce and be able to assimilate glucose. However, magnesium and zinc, other divalent cations, do not produce any effect, unlike in other cases where they can have activating and inhibitory effects [9,30]. This may indicate that the space for allosteric binding to cations is small since magnesium and zinc are larger ions. On the other hand, Tween 80 is a lipid solvent, so it can dissociate cell membranes. If it were a membrane β-glucosidase, alterations in its activity would be observed, but, as this is not the case, it functions as an enzyme that the yeast secretes to the outside and does not adhere to the membrane. In the case of mercury, it is known that it is generally an inhibitor since it acts on thiol groups of the catalytic domain or by binding to residues on the enzyme surface, changing its three-dimensional conformation [38].

The intracellular and extracellular β-glucosidases of *M. pulcherrima* are similar in pH, glucose tolerance, and interaction with certain metal ions such as magnesium or sodium, but both present many other differences [27]. Therefore, these are two different β-glucosidases and do not represent various forms produced by post-transcriptional processing, like the β-glucosidases of *Zygosaccharomyces bailii* [39].

### 3.7. Determination of Terpene Compounds Liberated from Wine Treated with Different M. pulcherrima Yeasts

The muscat juice was inoculated and fermented with a commercial *S. cerevisiae* strain. The physicochemical characteristics of the wine were: ethanol, 13.3% *v*/*v*; pH, 3.33; titratable acidity,4.6 g/L; volatile acidity, 0.35 g/L; and malic acid, 1.0 g/L. Afterward, Mp22, Mp24, and Mp26 isolates were inoculated (in triplicate assays) and the concentration of terpene compounds was determined (Table 3). The physicochemical characteristics of the wine were not significatively different from the original wine. No significant increase in the level of geraniol and nerol was observed after the addition of *M. pulcherrima* isolates. Mateo and Maicas [8] have previously proposed that both compounds could be produced by *Saccharomyces* strains during the alcoholic fermentation of Muscat wines. The concentration of oxide A and diol 2 were not increased despite the addition of *M. pulcherrima* isolates. All other oxides of linalool oxide B, oxide C, and oxide D) were not detected.

Concentrations of 4-vinylphenol, 2-methoxy-4-vinylphenol, and terpineol only increased after the addition of Mp22 or Mp24, but not when the Mp26 isolate was inoculated. Furthermore, 2-phenyl ethanol, linalool, and 4-vinylphenol content increased when *M. pulcherrima* isolates were added; these compounds are associated with fruity characteristics [8].

The aroma of varietal wines is determined by a wide variety of compounds, among which monoterpenes stand out. Most of these compounds are found forming glycosidic bonds, originating components that have no impact on the varietal aroma of wines. Therefore, their hydrolysis could enhance the varietal aroma of the wine. In the tests carried out in this study on wines of the muscatel variety, only a moderate increase in the concentration of terpenes has been obtained when treated with these isolates. However, if we take into account that the perception threshold of terpenes is very low [8], this increase is enough to increase the varietal aroma of this type of wine. These results are conditioned by the effect of various factors on glycolytic enzymes. The overall content of terpenic alcohols also seems to play an important role in the aromatic definition of wines from the Loureiro and Alvarinho varieties [40]. In addition, 2-Phenylethanol is also involved in conferring fruity and floral traits to wines, and its presence is related to the metabolic activity of non-*Saccharomyces* yeasts [41]. Our findings are similar to the observations of Fernandez-González et al. [42] who demonstrated the ability of various wine yeasts to hydrolyze the glycosylated terpene precursors, nor-isoprenoids. Winemaking with the addition of non-*Saccharomyces* strains has traditionally been associated with high concentrations of 4-vinylphenol and 4-vinyl-guaiacol that reach concentrations of up to 2 mg/L, producing olfactory characteristics reminiscent of different drugs [15]. The concentration of 4-vinylphenol in the analyzed wines was less than 1000 μg/L, which allows the use of our selected *M. pulcherrima* isolates in winemaking. Our results open up the possibility of using these yeasts to improve the aromatic characteristics of wines, regarding the release of terpenes.

## 4. Conclusions

Enzymes with β-glucosidase activity are of enormous interest in the field of applied microbiology. In the present work, an exocellular enzyme produced by *M. pulcherrima* has been characterized, showing some characteristics that make it very interesting for its industrial application, especially its tolerance to the presence of various sugars in the medium.

## Figures and Tables

**Figure 1 microorganisms-11-00964-f001:**
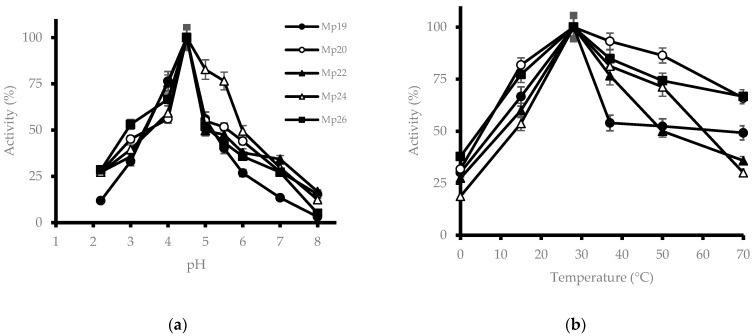
Effect of pH and temperature on β-glucosidase activity of different *M. pulcherrima* isolates. (**a**) pH; (**b**) temperature.

**Figure 2 microorganisms-11-00964-f002:**
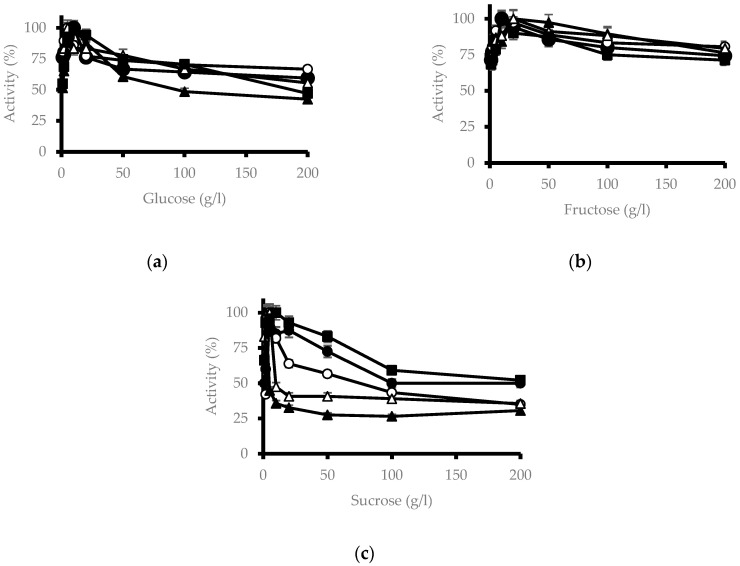
Effect of glucose, fructose, and sucrose on the β-glucosidase activity of different *M. pulcherrima* isolates, like Figure 1. (**a**) glucose; (**b**) fructose; (**c**) sucrose.

**Figure 3 microorganisms-11-00964-f003:**
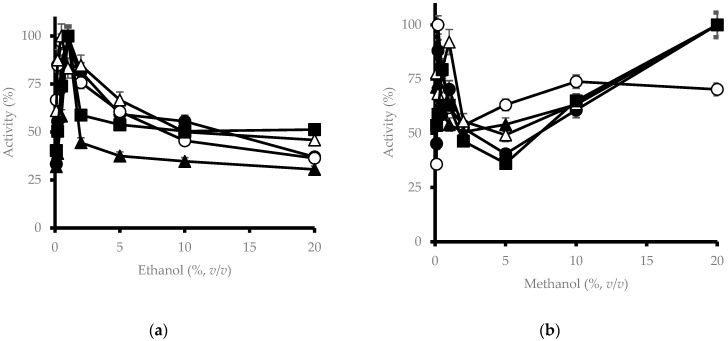
Effect of ethanol and methanol on the β-glucosidase activity of different *M. pulcherrima* isolates. Legend as Figure 1. (**a**) ethanol; (**b**) methanol.

**Table 1 microorganisms-11-00964-t001:** β-Glucosidase activity (expressed as absorbance at 404 nm) of the selected *M. pulcherrima* isolates.

Isolate	Absorbance
Mp19	0.253
Mp20	0.234
Mp22	0.31
Mp24	0.27
Mp26	0.256

**Table 2 microorganisms-11-00964-t002:** Effect of different cations and compounds on the β-glucosidase activity (expressed as absorbance at 404 nm) of the selected *M. pulcherrima* isolates.

				Isolate		
Compound	mM	Mp19	Mp20	Mp22	Mp24	Mp26
Ca^2+^	2	0.211	0.222	0.224	0.220	0.189
	10	0.502 *	0.501 *	0.537 *	0.475 *	0.485 *
Na^+^	2	0.212	0.205	0.205	0.225	0.199
	10	0.204	0.212	0.219	0.200	0.216
Mg^2+^	2	0.213	0.181	0.198	0.223	0.203
	10	0.229	0.180	0.222	0.219	0.226
Zn^2+^	2	0.205	0.190	0.221	0.206	0.205
	10	0.211	0.210	0.248	0.197	0.237
Tween 80	2	0.249	0.214	0.239	0.216	0.211
	10	0.234	0.261	0.246	0.252	0.239
Hg^2+^	2	0.233	0.233	0.282	0.231	0.236

* Statistically different (*p* < 0.05).

**Table 3 microorganisms-11-00964-t003:** Terpene compounds in Muscat wine (both control and inoculated samples). Concentration expressed as μg/L ^a^.

	Control	*M. Pulcherrima* Inoculated		
		Mp22	Mp24	Mp26
Oxide A ^b^	29.7 (1.2)	35.4 (2.1)	32.7 (3.2)	26.9 (3.4)
Oxide B ^c^	nd	nd	nd	nd
Linalool	20.0 (0.9)	50.4 * (3.9)	46.4 * (3.4)	39.2 * (5.3)
Ho-trienol	24.0 (3.2)	55.3 * (5.3)	39.1 * (4.2)	34.9 * (0.6)
2-Phenylethanol	1890.2 (43.4)	3117.5 * (39.8)	2817.8 * (26.8)	2638.5 * (45.6)
Oxide C ^d^	nd	nd	nd	nd
Oxide D ^e^	nd	nd	nd	nd
Terpineol	53.3 (3.4)	68.9 * (4.7)	66.2 * (1.2)	56.1 (3.9)
Nerol	24.6 (2.8)	24.8 (1.1)	22.4 (3.1)	25.3 (1.2)
Geraniol	59.8 (5.0)	60.9 (3.7)	57.9 (1.7)	58.2 (1.7)
Diol 1 ^f^	43.2 (4.7)	90.9 * (2.1)	88.2 * (2.1)	85.6 * (3.2)
4-Vinylphenol	63.2 (1.2)	86.3 * (2.4)	77.8 * (5.8)	63.1 (0.9)
Endiol ^g^	nd	51.2 * (2.1)	50.1 * (3.4)	33.7 * (4.2)
Diol 2 ^h^	12.0 (0.6)	11.4 (0.9)	9.8 (2.6)	11.1 (0.9)
2-Phenylethyl acetate	28.0 (4.1)	61.2 * (7.2)	33.3 (1.2)	35.8 (4.7)
2-Methoxy-4-vinylphenol	89.0 (6.1)	112.0 * (5.3)	116.4 * (6.5)	104.1 (2.9)

^a^ The values in brackets represent standard deviation (*n* = 3). ANOVA one factor, the significant difference is indicated as * (*p* < 0.05). ^b^ *cis*-5-vinyltetrahydro-1,1,5-trimethyl-2-furanmethanol. ^c^ *trans*-5-vinyltetrahydro-1,1,5-trimethyl-2-furanmethanol. ^d^ *cis*-6-vinyltetrahydro-2,2,6-trimethyl-2H-pyran-3-ol. ^e^ *trans*-6-vinyltetrahydro-2,2,6-trimethyl-2H-pyran-3-ol. ^f^ 2,6-Dimethyl-3,7-octadien-2,6-diol. ^g^ 2,6-Dimethyl-7-octene-2,6-diol. ^h^ 2,6-Dimethyl-2,7-octadien-1,6-diol. nd: not detected.

## Data Availability

No applicable.

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
