# Peer review of "Physico-Chemical Characterization of an Exocellular Sugars Tolerant Β-Glucosidase from Grape Metschnikowia pulcherrima Isolates"

_microorganisms, 2023, doi:10.3390/microorganisms11040964_

Round 1

Reviewer 1 Report

The manuscript of Tolosa entitled Physico-chemical characterization of an exocellular sugars tolerant Β-glucosidase from grape Metschnikowia pulcherrima isolates, reports the characteristics of a β- glucosidase enzyme activity in 5 strains of Metschnikowia pulcherrima (Mp19, Mp20, Mp22, Mp24, Mp26), which showed maximum activity at 28°C and pH 4.5, glucose and fructose tolerance, and lesser ethanol tolerance. The impact on winemaking was evaluated on a Muscat wine fermented with S. cerevisiae under winery conditions, and later, 3 of the isolates, Mp22, Mp24 and Mp26, were separately added to the wine. The concentration of volatile compounds shows small variation among isolates, but no significant effect on geraniol and nerol concentrations was detected. Authors concluded that some characteristics of exocellular enzyme produced by M. pulcherrima make it of some interest for industrial application. This manuscript is scientifically correct, and generally well written. However, the manuscript would benefit if the author could be more concise and focused on the results obtained and contrast them with others that have been already published on this topic, fitting together both sections in only one section “Results and Discussion”. Thus, overlapping could be avoided, as example in lines 201-214 results are discussed, comparing what obtained happened in this work with the results obtained by others (reference 8).  Contrasting the results obtained herein with those prior obtained by others, could highlight the outcomes of this work and thus conclusions could be improved.

Author Response

First of all, I want to thank you for your comments and the time spent reviewing my article. As reviewer 1 indicated, Results” and “Discussion” sections have been fitted together in only one section “Results and Discussion”. On the other hand, since a minimum number of 4000 words is established, it has been impossible to be more concise in the writing of the work.

Reviewer 2 Report

The manuscript characterized an exocellular enzyme produced by M. pulcherrima, which showed some features that make it very interesting for industrial applications, in particular its tolerance to the presence of various sugars in the medium. However, the author should consider the following aspects to improve their work.

1. In section 3.5, the reason for the sharp decrease in glucosidase activity is not explained.

2. The composition is not beautiful. It is recommended to reduce the size of graphical symbols. It is best to place the legend in the picture rather than in the caption.

3. The authors should write according to grammar rules. For example:

Line 8, there is no subject in the first sentence of abstract;

Line 11, the synthesis of an exocellular β-glucosidase has been demonstrated ane the optimal conditions to maximize the enzyme effectiveness were determined;

Line 109, the wine was clarified by centrifugation Samples were collected and and then with;

Line 118-120......

4. The results of the inoculation of M. pulcherrima strain to the Muscat wine after alcoholic fermentation is interesting. This work may be improved if the authors can consider the following points:

Its better to provide the physicochemical indexes, especially the total acid, volatile acid and pH, of the wines before and after the inoculation of M. pulcherrima, in order to evaluate the effect of M. pulcherrima on the wine solution.

Its better to provide the sensory analysis results, especilly the aroma characteristic and intensity, of the finished wines if the authors have the data, in order to evaluate the effect of M. pulcherrima on the global sensory features. According to the data in table 3, it is expected that the floral and fruity feature may be remarkably strengthened.

It is better to provide more results of the volatile components, for example the esters, in order to evaluate the effect of M. pulcherrima on important fermentation compounds inducing floral and fruity aroma. And the total terpenes may be summed in table 3.

If the M. pulcherrima strains behave well for all these aspects, the authors may further consider another question, whether the terpenes released from terpene glycosides can be held in the wine solution, which is a dynamic problem and papers have well discussed it (e.g., Xing-jie Wang. Phenolic matrix effect on aroma formation of terpenes during simulated wine fermentation - Part I: Phenolic acids. Food Chemistry, 2021, 341, 128288.). Of course, this is the scope of another article.

Author Response

First of all, I want to thank you for your comments and the time spent reviewing my article. Regarding the specific comments:

1.- The wording of the paragraph has been modified to improve its understanding

2.- The size of the symbols in the graphics has been reduced. Regarding the placement of the legend, it has been placed in the image

3.- The indicated modifications have been made

4.-

- The physical chemical data of the wine have been included

- No standard sensory analysis has been performed. The data is based on appraisals among laboratory personnel, so significant data cannot be provided.

- Since the objective of the work was to study the influence of the enzyme on the glycosylated precursors, it did not seem appropriate to carry out a parallel study on the ester content. They are typically fermentative compounds and are not related to the activity studied. Taking into account the ethanol content of the wine, it is difficult to think that the metabolic activity of M. pulcherrima is significant. On the other hand, other authors have observed an impact on the ester content when the musts are co-inoculated with M. pulcherrima and S. cerevisiae (Vaquero C, Loira I, Heras JM, Carrau F, González C, Morata A. Biocompatibility in Ternary Fermentations With Lachancea thermotolerans, Other Non-Saccharomyces and Saccharomyces cerevisiae to Control pH and Improve the Sensory Profile of Wines From Warm Areas. Front Microbiol. 2021 Apr 29;12:656262. doi: 10.3389/fmicb.2021.656262. PMID: 33995319; PMCID: PMC8117230).

- We are very grateful for the suggestion. I assure you that we will keep it in mind for future work.

Round 2

Reviewer 2 Report

It may be accepted after minor revision.

1) From the title of 2.9 and 3.7, the author concerns “volatile compounds liberated from wine” to evaluate the effect of M. pulcherrima yeasts. But the data the author provided was only terpene compounds. Terpene compounds are only a small part of volatile compounds in wine.

2) A titratable acidity of 3.6 g/L seems hard to support a pH value of 3.33. The author should re-check the data.

3) English checking is still required.

Author Response

Response to Reviewer 2

First of all, I want to thank you for your comments and the time spent reviewing my article. Regarding the specific comments:

1.- Titles from sections 2.9 and 3.7 have been modified

2.- Titratable acidity was 4.6 instead of 3.6 g/l

3.- Englis have been revised by a native colleague and no changes have been suggested.